# Coimmunization with Preerythrocytic Antigens alongside Circumsporozoite Protein Can Enhance Sterile Protection against *Plasmodium* Sporozoite Infection

Vladimir Vigdorovich,[a] Hardik Patel,[a] Alexander Watson,[a] Andrew Raappana,[a] Laura Reynolds,[a] William Selman,[a] Suzannah Beeman,[a] Paul T. Edlefsen,[b] Stefan H. I. Kappe,[a,c,d] D. Noah Sather[a,c,d]

aCenter for Global Infectious Disease Research, Seattle Children's Research Institute, Seattle, Washington, USA
bVaccine and Infectious Disease Division, Fred Hutchinson Cancer Research Center, Seattle, Washington, USA
cDepartment of Pediatrics, University of Washington, Seattle, Washington, USA
dDepartment of Global Health, University of Washington, Seattle, Washington, USA

Vladimir Vigdorovich and Hardik Patel made equal contributions. Author order was determined by consensus.

**ABSTRACT** Malaria-causing *Plasmodium* parasites have a complex life cycle and present numerous antigen targets that may contribute to protective immune responses. The currently recommended vaccine—RTS,S—functions by targeting the *Plasmodium falciparum* circumsporozoite protein (CSP), which is the most abundant surface protein of the sporozoite form responsible for initiating infection of the human host. Despite showing only moderate efficacy, RTS,S has established a strong foundation for the development of next-generation subunit vaccines. Our previous work characterizing the sporozoite surface proteome identified additional non-CSP antigens that may be useful as immunogens individually or in combination with CSP. In this study, we examined eight such antigens using the rodent malaria parasite *Plasmodium yoelii* as a model system. We demonstrate that despite conferring weak protection individually, coimmunizing each of several of these antigens alongside CSP could significantly enhance the sterile protection achieved by CSP immunization alone. Thus, our work provides compelling evidence that a multiantigen preerythrocytic vaccine approach may enhance protection compared to CSP-only vaccines. This lays the groundwork for further studies aimed at testing the identified antigen combinations in human vaccination trials that assess efficacy with controlled human malaria infection.

**IMPORTANCE** The currently approved malaria vaccine targets a single parasite protein (CSP) and results in only partial protection. We tested several additional vaccine targets in combination with CSP to identify those that could enhance protection from infection upon challenge in the mouse malaria model. In identifying several such enhancing vaccine targets, our work indicates that a multiprotein immunization approach may be a promising avenue to achieving higher levels of protection from infection. Our work identified several candidate leads for follow-up in the models relevant for human malaria and provides an experimental framework for efficiently carrying out such screens for other combinations of vaccine targets.

**KEYWORDS** *Plasmodium*, malaria, vaccine

Since the year 2000, the world has observed a steady decline in the global burden of malaria, but this trend has plateaued for the past 6 years, hinting at the insufficiency of current global efforts for malaria eradication. In 2020, the COVID-19 pandemic caused logistical interruptions in malaria prevention and treatment programs, resulting in a significant increase in global malaria burden, with an estimated 241 million new cases and 627,000 deaths, 77% of which were children under 5 years of age

Address correspondence to D. Noah Sather, noah.sather@seattlechildrens.org, or Stefan H. I. Kappe, stefan.kappe@seattlechildrens.org.

The authors declare no conflict of interest.

(1). It is clear that current interventions, including distribution of insecticide-treated bed nets for mosquito control and the use of several classes of drugs for the treatment of infection, are susceptible to disruption due to numerous factors, such as political instability, resistance to insecticides, and emerging resistance to antimalarial drugs. Further, the slowing gains in eradication outcomes prior to the pandemic indicate that these efforts may not be sufficient to achieve eradication without the addition of new prevention measures. Thus, although significant progress in malaria control and patient survival following infection has taken place since the turn of this century, achieving the future eradication of malaria will likely rely on the ongoing work to develop highly effective malaria vaccines.

Malaria disease is caused by multiple species of obligate intracellular parasites belonging to the *Plasmodium* genus. The sporozoite forms of the parasite are inoculated into the skin of a mammalian host by an infected *Anopheles* mosquito. Sporozoites then traverse the skin tissues in search of a blood vessel that they may invade to be carried with the blood circulation to the liver. Once in the liver, the sporozoites exit the circulatory system and, using unknown means, choose hepatocytes to invade and establish infections. In the course of liver infection, each of the 10 to 1,000 sporozoites (2) deposited by a mosquito bite can establish a liver stage, each producing 10,000 or more merozoite forms (3, 4) that exit the liver and initiate the blood-stage infection cycle in the host erythrocytes. Thus, targeting the relatively low-abundance sporozoite form of the parasite before the dramatic expansion of the parasite mass in the liver presents a critical opportunity for malaria vaccine design. Importantly, prevention of infection at this stage would stop both clinical disease and transmission.

The most advanced malaria vaccine—RTS,S/AS01—is based on a recombinant version of the circumsporozoite surface protein (CSP), which is highly abundant on the sporozoite surface. The WHO has recommended RTS,S/AS01 for use in young children because it has been shown to decrease incidence of severe malaria by ~30% (1, 5, 6). A similar subunit vaccine—R21/Matrix-M—has recently undergone a smaller phase 2b trial and yielded high antibody titers with a higher vaccine efficacy, compared to RTS,S/AS01, against clinical malaria (7, 8). Multiple studies strongly associated the magnitude and quality of antibody response with protection elicited by these vaccines (7, 9–12). The R21/Matrix-M study also reported that although antibody titers wane within a year, they could be boosted to near-initial levels (7) while maintaining high efficacy (8). Thus, CSP-based subunit vaccines have established a foundation for the development of next-generation malaria vaccines that may be achievable by including additional antigens from the *Plasmodium* parasite.

Recent work has shown the association of protection with responses to non-CSP antigens (13). When applied in the vaccine context, whole-sporozoite vaccine (WSV)-based approaches have shown that immunization with radiation- or chemically attenuated sporozoites can induce antibody responses against a broad spectrum of *Plasmodium falciparum* antigens, including CSP (14–16). Notably, the WSV-induced antibodies to non-CSP sporozoite antigens have been shown to inhibit sporozoite invasion of hepatocytes, implying that such antigens may elicit protective immune responses against preerythrocytic (PE)-stage infection (17). Similar vaccine-elicited sporozoite inhibition was recently shown more directly in the *Plasmodium yoelii* rodent malaria model by active coimmunization with CSP alongside a panel of PE antigens normally expressed during liver-stage development (18). However, protection in those experiments was assessed using an intravenous (i.v.) challenge, which, unfortunately, can miss protective mechanisms of antibodies active in the skin (19). Coimmunization of CSP and PE antigens in human clinical trials has been reported only for thrombospondin-related anonymous protein (TRAP) and RTS,S, although this study reported that coimmunization was actually detrimental to protection (20).

Previously, we used mass spectrometry to identify surface-exposed proteins in different *Plasmodium* species (21–23). Among the hits were proteins that localize to the sporozoite surface or those found in the invasion organelles (micronemes and

**TABLE 1** Constructs

| Construct name | Parasite localization | Mutations | UniProt reference | Domain | Amino acids |
|---|---|---|---|---|---|
| PyCSP | Surface | S27A, T348A[a] | A0A077Y0S6 | Ectodomain | 21–362 |
| PyGAMA | Micronemal | NA[b] | A0A077Y6C3 | Ectodomain | 22–603 |
| PySSP3 | Surface | NA | A0A077YBH7 | Ectodomain | 23–387 |
| PyTRAP | Micronemal | NA | A0A077YAH0 | Ectodomain | 23–752 |
| PyTRSP | Surface | NA | Q6IMC2 | Ectodomain | 21–145 |
| PyP52 | Micronemal | NA | A0A077YGW1 | Ectodomain | 24–458 |
| PyP36 | Micronemal | NA | A0A077YEG1 | Ectodomain | 71–356 |
| PyCelTOS | Micronemal | NA | Q6T944 | Ectodomain | 25–185 |
| PyHSP70-2 | ER, surface | NA | A0A077Y5M2 | Ectodomain | 22–651 |
| HIV-1 Env gp120 | NA | NA | P19550 | gp120 | 30–492 |

[a]PyCSP mutations that disrupt the N-glycosylation motifs.
[b]NA, not applicable.

rhoptries), ready to be released due to specific stimuli associated with the initiation of invasion. In addition, we showed that many components of the inner membrane complex could become exposed during the initial process of invasion and be accessible for recognition by antibodies (23). Recently, we demonstrated that passively transferred monoclonal antibodies (MAbs) raised against TRAP can enhance the protective effect of anti-CSP MAbs (24). In the present study, we investigated whether active immunization with non-CSP sporozoite antigens could enhance the anti-CSP response-mediated protection using the rodent parasite model. To achieve this, we developed a platform for evaluating synergistic antigen combinations that could potentially provide guidance for human malaria vaccine development. We screened recombinant versions of eight *P. yoelii* sporozoite-expressed proteins as PE immunogens in pairwise combination with CSP to test their ability to enhance the CSP-mediated sterile protection in mice upon mosquito-bite challenge. Overall, we identified four antigens, which, when used in combination with CSP, enhanced sterile protection over CSP alone, providing new strategies for the future development of effective multiantigen vaccines to prevent malaria infection.

## RESULTS

**Single-antigen vaccination with recombinant *P. yoelii* sporozoite antigens elicits variable degrees of infection blocking despite strong antibody responses.** To establish the potential efficacy of single-protein vaccine formulations against liver infection, we selected ectodomains of nine PE vaccine antigens expressed by *P. yoelii* sporozoites for recombinant production and purification alongside an unrelated control protein, HIV-1 Env gp120 (Table 1 and Fig. 1A). Constructs encoding ectodomains of *P. yoelii* proteins—circumsporozoite protein (PyCSP), glycosylphosphatidylinositol (GPI)-anchored micronemal antigen (PyGAMA), sporozoite surface protein 3 (PySSP3), thrombospondin-related anonymous protein (PyTRAP), thrombospondin-related sporozoite protein (PyTRSP), P36 (PyP36), P52 (PyP52), cell-traversal protein for ookinetes and sporozoites (PyCelTOS), and heat shock protein 70-2 (PyHSP70-2)—were engineered for production in HEK293 cells. Additionally, the PyCSP construct contained two point mutations to disrupt the N-linked glycosylation motif and prevent this posttranslational modification. Groups of BALB/cJ mice ($n = 3$) were primed with 20 $\mu$g of individual *P. yoelii* sporozoite antigen in 20% Adjuplex adjuvant and 14 days later received an identical booster dose (Fig. 1B). Adjuplex is an experimental adjuvant that is a biodegradable matrix of carbomer homopolymer (Carbopol) and nanoliposomes derived from soy lecithin (25) and is known to stimulate robust antibody responses in recombinant protein vaccine formulations. Carbomer-based compounds (acrylic acid polymers) are under investigation as synthetic adjuvants, due to their ability to stimulate robust immunity and inherent low toxicity (25). They have been extensively studied in many preclinical models for vaccines against numerous pathogens, including parasites, viruses, and fungi (26–29).

Recognition of recombinant and sporozoite-derived antigens by vaccine-elicited antibodies was assessed using plasma samples collected prior to the mosquito-bite

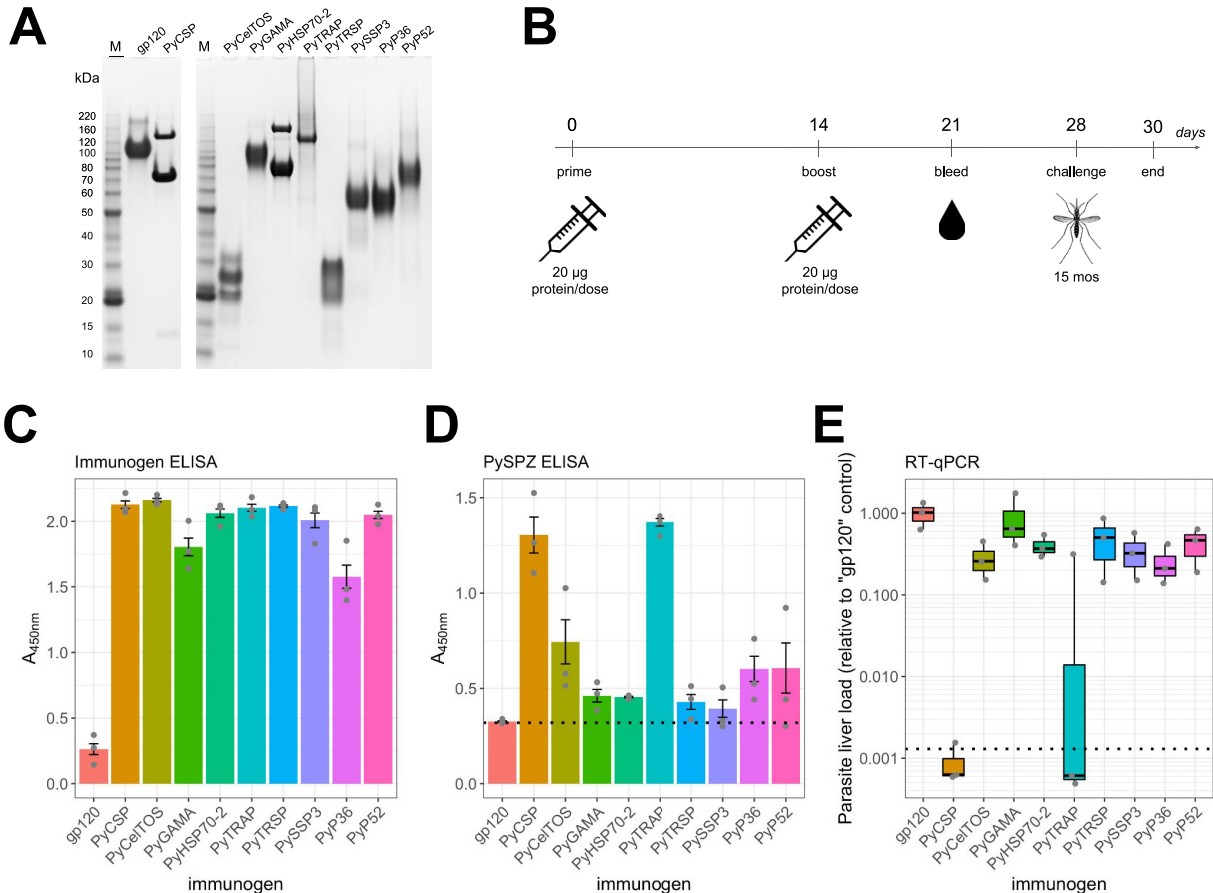

**FIG 1** Immunization with recombinant *P. yoelii* PE antigens induces specific antibody responses but various degrees of protection against infection in the liver. (A) SDS-PAGE analysis of the nine recombinant *P. yoelii* PE antigens used in immunizations and a control protein, HIV-1 Env gp120 ("gp120"). (B) The immunization-challenge strategy schematic. Ten groups of mice (*n* = 3) were immunized twice with each of the PE antigens (in 20% Adjuplex) 14 days apart, followed by blood sample collection on day 21, and challenged with bites from 15 *P. yoelii*-infected mosquitoes on day 28. Parasite liver load was determined by RT-qPCR after harvesting the livers on day 30 (45 h postchallenge). (C and D) The bar graphs show total antibody reactivity in 1:50-diluted "Day-21" plasma to purified PE antigens (C: each immunization group was tested against its corresponding antigen) or to whole sporozoites (D) by ELISA, with bar heights corresponding to the mean signal and the error bars representing ±standard errors of the means (SEM). In panel D, the empirical background level for the sporozoite samples is indicated by a dotted line corresponding to reactivity against the control ("gp120") protein. (E) The parasite liver load (*P. yoelii* 18S rRNA) quantified by RT-qPCR. Each mouse is represented by a gray point showing relative values of parasite liver load normalized with the mean value from the "gp120" control group. The box plot displays the upper and lower quartiles, median value, and whiskers representing the data range. The dotted line denotes the limit of detection, defined as the mean signal from the no-template control.

challenge (on day 21; i.e., 1 week postboost). Plasma from each animal exhibited strong recognition of its respective recombinant immunogens by enzyme-linked immunosorbent assay (ELISA) at a dilution of 1:50, with the exception of the animals immunized using the gp120 control protein (which showed poor immunogenicity, based on these data), suggesting that all recombinant PE immunogens elicited a robust antibody response (Fig. 1C). We next tested the recognition of natively expressed antigens by performing ELISA with plated whole wild-type *P. yoelii* sporozoites to assess the antigenic similarity between recombinant and natively expressed antigens. Our results revealed various degrees of antibody reactivity (Fig. 1D) with respect to the background signal defined by the gp120 control condition (this protein is not expressed by the sporozoites), showing strongest reactivity in PyCSP- and PyTRAP-immunized plasma; moderate reactivity in plasma from PyCelTOS, PyP36, and PyP52 immunizations; and lower reactivity in plasma from the PyGAMA, PyHSP70-2, PyTRSP, and PySSP3 immunizations. Despite the differences in the magnitude of signal in whole-sporozoite recognition, these results indicated that the antibodies elicited by the PE immunogens could recognize both recombinant and sporozoite-associated antigens, indicating a degree of antigenic similarity between recombinant and natively expressed antigens.

Following the immunization, mice were challenged with bites from 15 *P. yoelii*-infected mosquitoes (on day 28), and the parasite liver-stage infection biomass was assessed by *P. yoelii*-specific 18S reverse transcriptase quantitative PCR (RT-qPCR) relative to the control group immunized with the HIV-1 Env gp120 subunit. Mice immunized with single PE antigens exhibited various degrees of protection from infection, manifested in a reduction in the parasite liver-stage burden among the groups after challenge. The mice immunized with PyCSP or PyTRAP showed the most inhibition of infection, showing the lowest liver load, below the detection limit of the *P. yoelii* 18S RT-qPCR assay, representing >1,000-fold reduction compared to the "gp120" control (Fig. 1E). No other single-antigen vaccination regimen induced responses that reduced liver infection below the detection threshold. However, the remaining groups demonstrated reduction in parasite liver load ranging from 1.5- to 4.8-fold compared to the control group, indicating that liver infection was blocked to various degrees. These results suggest that immunization with PyCSP or PyTRAP induces strong blocking antibodies against *P. yoelii* sporozoite infection, whereas the rest of the *P. yoelii* PE antigens elicited antibodies that were less inhibitory.

Notably, when accounting for the decrease in liver-stage burden (Fig. 1E), our plasma reactivity data imply that immunization groups with the highest signal against whole *P. yoelii* sporozoites (PyCSP and TRAP) also demonstrate the greatest protective effect against sporozoite infection. This may reflect differences in relative abundance (30) and/or accessibility of the target antigens at the sporozoite stage or be due to nuances in their biological function and susceptibility to antibody inhibition. Regardless, these collective findings indicate that immunization with recombinant PE antigens induced strong antibody titers that recognize their cognate sporozoite-associated protein and block liver infection, albeit to various degrees. However, the inhibition of sporozoite infection achieved by these PE immunogens (with the exception of TRAP) is moderate compared to CSP-mediated inhibition, indicating that they are likely insufficient as standalone vaccines.

**Combined immunization of PE antigens with PyCSP identifies protective and nonprotective coimmunogens.** We next examined whether any of the PE immunogens could enhance PyCSP-mediated sterile protection in coimmunization/challenge experiments. Our standard immunization regimen to study sterile protection uses three doses of 20 $\mu$g of PyCSP in BALB/cJ mice and induces a strong protective effect during the mosquito-bite challenge (see Fig. S1A in the supplemental material), which is too potent to detect enhancement of sterile protection by non-CSP antibodies. Thus, we evaluated several PyCSP immunization regimens to identify conditions resulting in sterile protection of 10 to 50% of mice per group, which would allow us to detect even nuanced statistically significant improvements in protection due to the addition of PE immunogens. In a pilot experiment, we observed that a 2-dose regimen with 5 $\mu$g of PyCSP induces 20% sterile protection (Fig. S1B). Therefore, we chose the regimen consisting of 7 $\mu$g PyCSP combined with 13 $\mu$g PE immunogens or the control gp120 antigen (for a 20-$\mu$g total protein dose per injection in 20% Adjuplex) as our standard coimmunization strategy to assess changes in vaccine efficacy. Mice were immunized twice with combinations containing each single PE immunogen (or gp120 control) with PyCSP or gp120-only control, 2 weeks apart, and then challenged with bites from 15 *P. yoelii*-infected mosquitoes 14 days postboost (Fig. 2A). Using this regimen, the mice immunized with gp120 + PyCSP ($n = 30$) showed 20% sterile protection (Fig. 2B), compared to the gp120-alone group (no PyCSP, $n = 30$) showing 0% sterile protection ($P = 0.013$, Barnard's exact test; Table 2). Importantly, we observed that these control groups produced consistent outcomes over multiple experiments (Table S1).

Among the PE antigen/PyCSP combinations tested, we identified three groups that showed significantly enhanced sterile protection over control: PyCelTOS (53%, $P = 0.026$), PyHSP70-2 (50%, $P = 0.027$), and PyTRAP (50%, $P = 0.027$). Coimmunization with PyP52 resulted in 47% sterile protection, which was not significant ($P = 0.075$) (Table 2). When data were analyzed to account for the delay to onset of blood-stage patency among the groups using the log rank test, PyCelTOS ($P = 0.014$), PyHSP70-2 ($P = 0.032$), and PyTRAP ($P = 0.004$) again showed statistical significance; however, here

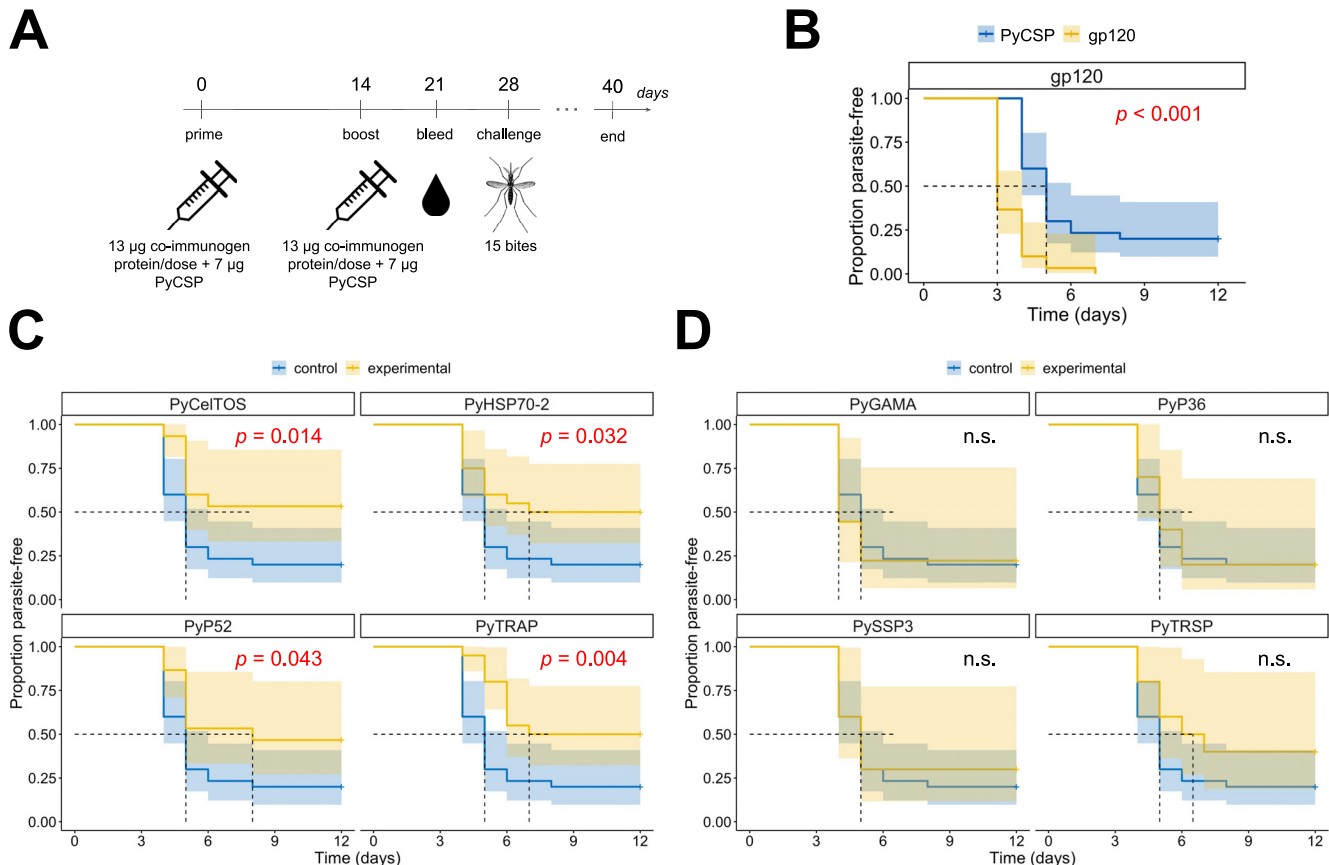

**FIG 2** Coimmunization of PyCSP with recombinant sporozoite proteins delineates the protective and nonprotective coimmunogens. (A) Groups of mice were immunized with a mix of 7 µg PyCSP and 13 µg of another PE antigen (as listed in Table 2 and Table S1 in the supplemental material) following the same immunization-challenge strategy described for Fig. 1B. The level of sterile protection was determined by screening Giemsa-stained blood smears for parasitemia on days 3 to 12 postchallenge. (B) The "gp120+PyCSP" group (blue line, n = 30) showed the induction of significant sterile protection (20%), compared to the "gp120" control group (yellow line, n = 30, 0%), establishing the baseline of protection induced by PyCSP. (C) Coimmunizations with PyTRAP (n = 20), PyHSP70-2 (n = 15), PyCelTOS (n = 20), and PyP52 (n = 15), shown as a yellow line in each panel, demonstrate enhanced protection over the "gp120+PyCSP" group (blue line, same data as in panel B). (D) Coimmunizations with PyTRSP (n = 10), PyGAMA (n = 9), PyP36 (n = 10), and PySSP3 (n = 10), shown as a yellow line in each panel, were not statistically distinguishable from the "gp120+PyCSP" control group (blue line). The log rank test was used to assess statistical significance of difference in each Kaplan-Meier plot, with P values above 0.05 represented by "n.s."

PyP52 ($P = 0.043$) also showed statistically significant improvement over the control gp120/PyCSP group (Fig. 2C). The remaining PE antigen/PyCSP groups yielded lower proportions of sterilely protected animals following the challenge and were not otherwise statistically distinguishable from the control group (Table 2 and Fig. 2D). Thus, of the 8 PE antigens tested, four improved sterile efficacy over the PyCSP immunization condition: PyCelTOS, PyTRAP, PyHSP70-2, and PyP52.

**TABLE 2** Challenge outcomes following coimmunization (Barnard's exact test)

| Group | Immunogen 1 (13 µg) | Immunogen 2 (7 µg) | No. infected | Total no. | Protection ratio (%) | P value vs group A[a] | P value vs group B[a] |
|---|---|---|---|---|---|---|---|
| A | gp120 | NA[b] | 30 | 30 | 0 | NA | NA |
| B | gp120 | PyCSP | 24 | 30 | 20 | **0.013** | NA |
| C | PyHSP70-2 | PyCSP | 10 | 20 | 50 | NA | **0.027** |
| D | PyTRAP | PyCSP | 10 | 20 | 50 | NA | **0.027** |
| E | PyCelTOS | PyCSP | 7 | 15 | 53 | NA | **0.026** |
| F | PyP52 | PyCSP | 8 | 15 | 47 | NA | 0.075 |
| G | PyP36 | PyCSP | 8 | 10 | 20 | NA | 1.000 |
| H | PyGAMA | PyCSP | 7 | 9 | 22 | NA | 0.923 |
| I | PySSP3 | PyCSP | 7 | 10 | 30 | NA | 0.585 |
| J | PyTRSP | PyCSP | 6 | 10 | 40 | NA | 0.239 |

[a]Unadjusted Barnard's exact test P value.
[b]NA, not applicable. Bold lettering was used for values that reached statistical significance ($P < 0.05$).

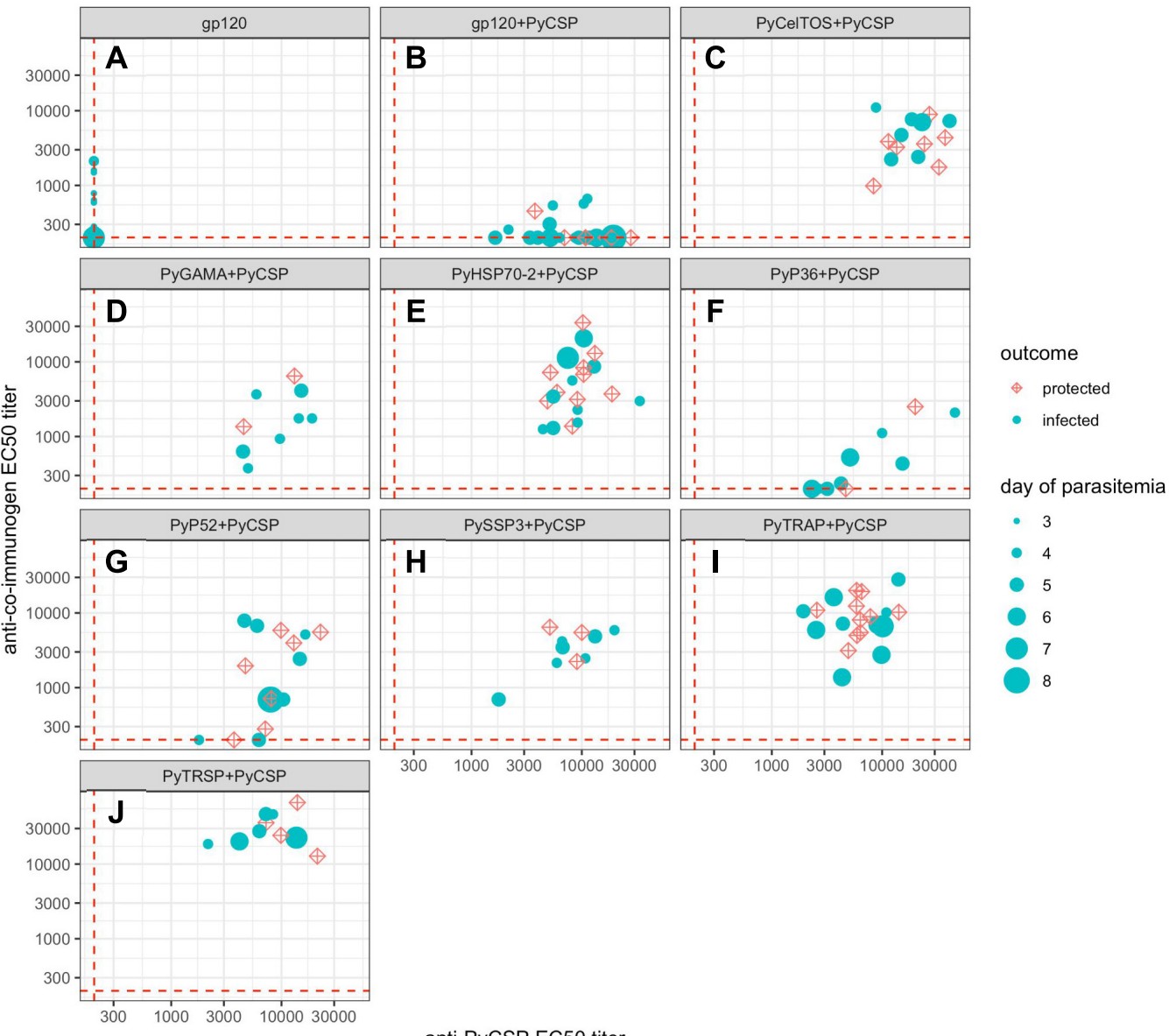

**FIG 3** Coimmunization of PyCSP with each of the candidate proteins leads to robust antibody responses. Total antibody titers represented by the $EC_{50}$ for each group were determined by ELISA using the "Day-21" plasma samples and plotted as the anti-PyCSP titer on *x* axes and the anticoimmunogen titer on *y* axes. Each data point represents an individual mouse that either became infected (shown as the cyan filled circle; the size of the circle suggests the day when parasitemia was first detected) or remained free of detectable parasites (represented by a red crossed-diamond symbol). The $EC_{50}$ limit of detection for all the antigens was set at 1:200 dilution (dashed red lines), and all the values below that threshold were adjusted to this value.

To quantitate the relative antibody response against each coimmunogen, and whether PE antigen titers were associated with the occurrence of sterile protection, we assessed the total anti-PE-antigen immunoglobulin levels in each of the coimmunization groups (Fig. 3 and Fig. S2). For the majority of PE antigens tested, each of the animals immunized had half-maximal effective concentration ($EC_{50}$) titers above 1:1,000, with PyTRSP yielding the highest $EC_{50}$ titers (1:10,000 to 1:70,000). Notably, PyP36 and PyP52 had significantly lower $EC_{50}$ titers with multiple animals measuring below the limit of detection (Fig. 3F and G). In addition, most mice immunized with the gp120 control protein had $EC_{50}$ titers below the limit of detection (Fig. 3A and B), indicating poor immunogenicity of gp120 in these mice. Importantly, our data showed that regardless of the coimmunogen used, PyCSP elicited comparable $EC_{50}$ titers across groups (1:1,000 to 1:45,000), demonstrating the independence of antibody responses

to PyCSP from those elicited by PE immunogens under the coimmunization conditions used here. Correlation analysis between the $EC_{50}$ titer and the protection status for each coimmunization group revealed no statistically significant correlations for either anti-CSP (Fig. S3A) or anti-PE antigen (Fig. S3B) response used in each respective group. Likewise, no statistically significant correlation was detected between antibody $EC_{50}$ titer and the day of blood-stage parasitemia detection (Fig. S4A and B). Taken together, our data indicate that the magnitude of antibody response alone is an insufficient predictor of either sterile protection status or a delay to blood-stage patency.

## DISCUSSION

The RTS,S vaccine experience suggests that increasing the protective efficacy of CSP-based PE malaria vaccines should be a major priority for the field as it enhances the value of such a vaccine for malaria eradication efforts. However, developing a CSP-only malaria vaccine with high efficacy and durability of protection has been hampered by various roadblocks, and identifying additional PE vaccine antigens to augment antisporozoite immunity may be a promising strategy to fortify CSP-based vaccines. In our recent work, we have shown that plasma from mice immunized with recombinant TRAP—an exemplar PE antigen—inhibited sporozoites and that passively transferred monoclonal antibodies to *P. falciparum* TRAP (PfTRAP) could enhance anti-PfCSP-mediated protection from mosquito-bite challenge (24). Others have shown that active immunization with TRAP-CSP fusion proteins had protective effects in a similar experiment using the *Plasmodium berghei*-infected mosquito-bite challenge, although the individual contributions of anti-TRAP and anti-CSP responses were not completely defined (31). Additionally, a broader screen of coimmunogens derived from antigens expressed during liver-stage development identified candidates that may be useful for eliciting T-cell responses (18). Here, we show that several recombinant PE antigens normally expressed in sporozoites, when used in two-protein coimmunizations alongside PyCSP, can enhance sterile protection afforded by PyCSP alone.

Only four of the eight PE coimmunogens tested showed a statistically significant enhancement of anti-PyCSP response-associated sterilizing protection, even though all eight candidates elicited strong binding antibody responses. These differences in protective outcomes may be explained by the cellular localization differences among these protein targets, or by the degree to which their respective biological functions can be disrupted by vaccine-elicited interventions. In particular, CSP, TRAP, TRSP, P52, P36, and CelTOS have been previously associated with invasion functions of the parasite (32–37). Previous work has also shown that immune response to CelTOS is associated with sterile protection from *Plasmodium berghei* challenge (38). HSP70-2 is an endoplasmic reticulum (ER)-resident chaperone that has been shown to associate with P36 and P52 (37). In contrast, neither GAMA nor SSP3 to date has been ascribed a role in PE stages. Nevertheless, in the context of vaccine targeting, disruption of protein function may not be required for protection, since antibody engagement of surface-exposed molecules alone may disrupt the ability of the parasite to traverse tissues (19) or mark the parasite for immune-mediated clearance (39).

Overall, our data show that the total polyclonal binding antibody titers resulting from immunization were not directly associated with sterile protection in this study. It is possible that protection is associated with specific functional antibody subsets within the polyclonal antibody mixture based on either on-target epitope specificity (40), antibody isotype (10, 11), or off-target cross-reactivity (41). We and others had previously shown that the composition of the polyclonal antibody response can be diverse enough to include antibodies capable of inhibiting the parasite as well as those that do not have a detectable impact on the course of infection despite their ability to bind the antigen (24, 42–46). If this is the case here, then a more extensive characterization of the antibody response with a focus on specific "vulnerable" epitopes would be required to describe the antibody response patterns that lead to protection. Further, it is probable that specific characteristics of the antibody response, such as avidity, antibody maturation,

and IgG subclass distribution, may be more directly associated with protection (47) and provide mechanistic insights into protective responses, and these characteristics will be an intense focus in our follow-up studies. However, we cannot rule out that the lack of correlation between antibody titers and protection status may be indicative of a more prominent role for cellular immunity, as has been observed in other similar systems (9, 48–50).

Our antibody binding data from the whole-sporozoite ELISAs imply that gross antigenic differences between the recombinant and sporozoite-associated proteins were unlikely to be the primary reason for lack of significant protection in single-protein immunizations. Despite the differences in glycosylation patterns of *Plasmodium* proteins (51), compared with the mammalian recombinant versions, our data showed that the elicited antibodies recognized both the recombinant immunogens and the native sporozoite-derived proteins. The variable signal strength in the whole-sporozoite ELISAs is likely indicative of different expression levels for each of these antigens in sporozoites (30), not of the strength of antibody response (the robustness of which is strongly indicated by the robust recognition of the recombinant versions). However, we cannot rule out the possibility that a subpopulation of antibodies may poorly recognize the native antigen, reducing the effective concentration of on-target circulating antibodies. A more extensive dissection of epitope targeting and the identification of neutralizing epitopes in these PE antigen targets will provide insight into this and would potentially enable the improvement of efficacy mediated by those antigens through iterative vaccine design.

To date, coformulation with RTS,S in a clinical setting against *P. falciparum* infection has been tested only using TRAP recombinant protein, and in that study the RTS,S/TRAP/AS02 combination failed to protect against a controlled human malaria infection (20) despite prior work showing protection for RTS,S/AS02 alone (52). This created the concern that PE subunit vaccine coimmunization may be detrimental to CSP-mediated vaccine protection. In contrast, mixed regimens of RTS,S/AS01B and a viral vector-encoded multiple-epitope TRAP (ME-TRAP) construct elicited significant sterile protection from mosquito-bite challenge, albeit with no significant difference from the control RTS,S/AS01B-alone group (53, 54). Our mouse challenge data suggest that, when used in single-antigen immunizations, PyTRAP resulted in blocking of liver infection comparable to that with PyCSP, with all other immunogens in our panel yielding weak blocking activity. When used in combination with PyCSP, PyTRAP coimmunizations consistently resulted in enhanced probability of sterile protection compared to PyCSP-alone immunizations. Taken together with our recent finding that anti-TRAP MAbs can increase chances of sterile protection afforded by anti-CSP MAbs in a humanized mouse system (24), these data suggest that the use of recombinant TRAP in coimmunization strategies still holds promise and should be revisited.

In summary, our study provides an early-stage proof-of-concept that immunization with recombinant CSP can be further enhanced by the addition of other sporozoite antigens, resulting in greater chances of sterile protection. All of the coimmunogens examined here have orthologs in *P. falciparum*. However, since our observations were made in a rodent malaria model, it is important moving forward to evaluate the efficacy of the *P. falciparum* versions in a physiologically relevant experimental system like the humanized-liver FRG-huHep mice (55). Some discrepancies have been observed previously between the rodent and human malaria models in vaccine development, and so confirmation of these findings against human-infective malaria is a critical next step. Additionally, it is likely that each antigen combination requires extensive optimization of dosages and formulations to enhance the relative antibody responses to achieve maximal efficacy, and careful evaluation of relevant epitope targets may allow rational design approaches. Future aspects of this study will also focus on testing the combinations of more than two antigens for achieving higher levels of sterile protection and determining the optimal combinations and formulations to advance into the clinical setting. Overall, our findings pave the way for the development of combinatorial

immunization regimens that may significantly enhance the protective properties of RTS, S and the next-generation subunit vaccines.

## MATERIALS AND METHODS

**Constructs and protein production.** Sequences encoding the ectodomains of *Plasmodium yoelii* proteins (Table 1) were codon optimized and cloned into the pcDNA-3.4 expression vector flanked by the tissue plasminogen activator (tPA) signal sequence on the 5′ end and the 8×His and AviTag sequences on the 3′ end. The two N-glycosylation sites present (identified by the NxS/T motif) in PyCSP were mutated to NxA (S27A, T348A), since it is currently unclear whether *Plasmodium* spp. carry out this protein posttranslational modification (56).

Recombinant proteins were produced using the suspension HEK293-based expression system, as previously reported (57). Briefly, cultures of FreeStyle 293 (Thermo) cells were transiently transfected with the constructs using PEI Max (Polysciences) at a 1:4 (wt/wt) DNA/polyethyleneimine (PEI) ratio and maintained for 5 days prior to harvesting the supernatants. Purification consisted of immobilized metal-affinity chromatography using HisPur resin (ThermoFisher) followed by gel filtration on a calibrated HiLoad Superdex 200-pg column (Cytiva). Protein was then concentrated and stored at 4°C or flash-frozen (and stored at −20°C) until use. The HIV-1 Env gp120 control protein was produced as previously described (58).

**Animal studies ethics statement.** All procedures involving animals were performed in adherence to protocols reviewed and approved by the Institutional Animal Care and Use Committee (IACUC) at the Seattle Children's Research Institute (protocol identifier [ID] IACUC00487).

**Immunization.** BALB/cJ mice (purchased from Jackson Laboratories, Bar Harbor, ME) were selected for this study, which enabled the use of our optimized *P. yoelii* immunization and mosquito-bite challenge model, which has been developed and validated by us previously (24, 47, 59). Mice were intramuscularly immunized twice—2 weeks apart—with 20-$\mu$g total protein doses containing 13 $\mu$g of the candidate immunogens or control (gp120) and 7 $\mu$g of PyCSP, or 20 $\mu$g control protein in 20% Adjuplex adjuvant (Empirion LLC, Columbus, OH). Blood samples from immunized animals were collected on day 21 (i.e., 1 week after the last immunization) through submandibular bleeding. The plasma was harvested by centrifuging blood samples at 4,500 × $g$ for 10 min at 4°C and then immediately stored at −20°C until further use for the assessment of antibody responses.

The experiments were performed in two stages. First, a pilot stage with 5 mice per immunogen group was used to approximate protection outcomes, which was used to calculate the minimal numbers of animals per group needed to reach statistical power, according to Barnard's exact test results (alpha = 0.05, beta = 0.2). In the second stage, several replicate experiments were performed to increase the group sizes and to capture the diversity of challenge efficiencies typically observed for multiple individual experiments.

**Mosquito-bite challenge and protection.** The mosquito-bite challenge was done as described previously (59). Briefly, the *Anopheles stephensi* mosquitoes were fed with *P. yoelii* 17XNL strain-infected blood meal and the infection rate was determined on day 9 after blood meal by analyzing mosquito midguts for the presence of oocyst. A small proportion of mosquitoes were dissected on day 14 after mosquito infection to verify the presence of sporozoites in the salivary glands to ensure the batch exceeded a minimum average cutoff of 10,000 parasites per mosquito. On day 15, immunized mice were anesthetized with ketamine (100 mg/kg body weight)-xylazine (10 mg/kg body weight) solution and placed above small cages each containing 15 mosquitoes, one mouse per cage. The mice were rotated for 10 min between the cages after every 50 s of incubation in order to minimize the variations in mouse infection and to maximize the probing events, as opposed to feeding. Sterile protection was determined by screening Giemsa-stained thin blood smears, made from mouse tail snip bleeding, between days 3 and 12 after mosquito-bite challenge. Mice were considered protected if blood-stage parasites were absent while examining at least 30 microscopic fields per mouse.

**Quantitation of antibody responses.** Total antibody responses were quantitated by direct-immobilization ELISA. Antigens were diluted in 0.1 M sodium bicarbonate buffer and incubated overnight at room temperature in wells of Immulon 2HB 96-well plates (Thermo Scientific) at a dose of 50 ng/well. Plates were then washed five times with phosphate-buffered saline (PBS) containing 0.02% Tween 20, which was performed between each step in the assay. Plates were then blocked against nonspecific binding with 10% nonfat milk and 0.3% Tween 20 diluted in PBS for 1 h at 37°C. Plasma samples obtained following immunization were heat inactivated at 56°C for 30 min and diluted in PBS with 10% nonfat milk and 0.03% Tween 20 for a range of 1:20 to 1:5,598,720. Following an incubation for 1 h at 37°C, the plates were washed and incubated in a 1:2,000 dilution of horseradish peroxidase (HRP) goat anti-mouse Ig (BD, catalog no. 554002) in PBS with 10% nonfat milk and 0.03% Tween 20. Plates were then incubated for 1 h at 37°C. Plates were developed using 50 $\mu$L/well of SureBlue Reserve tetramethylbenzidine (TMB) reagent (SeraCare Life Sciences Inc., catalog no. 5120-0083), and development was stopped after 3 min at room temperature by the addition of 50 $\mu$L/well of 1 N sulfuric acid. Absorbance readings at 450 nm were performed using an ELx800 microplate reader (BioTek), and raw data were used to generate titration curves with the R package drc (version 3.0-1) to estimate the $EC_{50}$ dilution values (sample data and fits are shown in Fig. S5 in the supplemental material).

**Antibody recognition of sporozoites in ELISA.** The *P. yoelii* 17XNL sporozoites were harvested from the salivary glands of infected *Anopheles stephensi* mosquitoes as described previously on day 15 after the infected blood meal (60). The sporozoites were diluted in phosphate-buffered saline (PBS; Corning catalog no. 21-040-CV) and deposited at 20,000 sporozoites/well in Immulon 2HB 96-well plates

(Thermo Scientific), followed by an incubation overnight at 4°C. Parasites were then fixed by air drying by aspirating all liquid from each well and incubating at room temperature for 25 min. Plates were then stored at −20°C until use. Following thawing, plates were blocked with 50 $\mu$L of 1% bovine serum albumin (BSA) (VWR, catalog no. 97061-422) in PBS for 1 h at room temperature. Plasma samples obtained following immunization were heat inactivated at 56°C for 30 min and diluted in PBS with 1% BSA to achieve a 1:25 dilution. Fifty microliters of diluted sample was then added to the 50-$\mu$L block buffer on the plate in duplicate to achieve a final dilution of 1:50 and incubated at room temperature for 2 h. Plates washed with 300 $\mu$L of PBS (pH 7.4) six times. Following washing, a 1:2,000 dilution of HRP goat anti-mouse Ig (BD, catalog no. 554002) diluted in PBS with 1% BSA was added to the plate and incubated for 1 h at room temperature. Plates were again washed with PBS as described above. Plates were developed using 50 $\mu$L/well of SureBlue Reserve TMB reagent (SeraCare Life Sciences Inc., catalog no. 5120-0083), and development was stopped after 3 min at room temperature by the addition of 50 $\mu$L/well of 1 N sulfuric acid. Absorbance readings at 450 nm were performed using an ELx800 microplate reader (BioTek).

**Determination of parasite liver load.** The parasite load in the liver was determined as described previously (61). Briefly, the livers were perfused 45 h after mosquito-bite challenge with 1× PBS, and total RNA was extracted from the lower right caudate process of the caudate lobe of the liver from each mouse using the miRNeasy kit (Qiagen; 217004). The cDNA was synthesized from 1 $\mu$g of total RNA using the QuantiTect reverse transcription kit (Qiagen; 205311), and quantitative PCR (qPCR) was performed using Bimake SYBR green Master Mix (catalog no. B21202) on a QuantStudio 5 real-time PCR system. Briefly, 0.5 $\mu$L of diluted cDNA (1:10) was used in a total 10-$\mu$L reaction volume to amplify *P. yoelii* 18S rRNA using 5′-GGGGATTGGTTTTGACGTTTT-3′ (forward primer) and 5′-AAGCATTAAATAAAGCGA ATA-3′ (reverse primer) and the mouse *Gapdh* gene, as an internal control, using 5′-CCTCAACTACAT GGTCTACAT-3′ (forward primer) and 5′-GCTCCTGGAAGATGGTGATG-3′ (reverse primer). The parasite liver load was calculated by comparative threshold cycle ($C_T$) analysis and represented as $2^{(-\Delta\Delta C_T)}$ in relation to the gp120 control group (62). The limit of detection was determined using a negative-control reaction mixture where no template cDNA was added.

**Data processing and statistics.** Statistical tests were nonparametric (Barnard's, log rank) and employed with standard sidedness and an 0.05 unadjusted *P* value threshold. Due to small sample sizes, *P* values were not adjusted for multiple comparisons. Raw data were processed using R (version 4.0.2) and packages tidyverse (version 1.3.1), drc (version 3.0-1), Exact (version 3.0), survival (version 3.2-13), and rstatix (version 0.7.0). Plots were generated using packages ggplot2 (version 3.3.5), ggpubr (version 0.4.0.999), and survminer (version 0.4.9).

**Data availability.** All data are available in the article and supplemental material.

## SUPPLEMENTAL MATERIAL

Supplemental material is available online only.
**SUPPLEMENTAL FILE 1**, PDF file, 0.7 MB.

## ACKNOWLEDGMENTS

We thank the staff of the vivarium at Seattle Children's Research Institute for their support of the animal studies presented here. In addition, we thank Nana Minkah for his help with the study design and Tess Seltzer (of the insectary staff) for her diligent work in rearing the mosquitoes for these studies.

This work was funded by NIH R01 AI117234 to D.N.S. and S.H.I.K.

Conceptualization and experimental design, V.V., H.P., S.H.I.K., and D.N.S.; investigation: V.V., H.P., A.W., A.R., L.R., W.S., and S.B.; data analysis and visualization, V.V., H.P., P.T.E., S.H.I.K., and D.N.S.; writing—original draft, V.V.; writing—review and editing, V.V., H.P., P.T.E., S.H.I.K., and D.N.S.; resources, S.H.I.K. and D.N.S.; supervision, project administration, and funding acquisition, V.V., S.H.I.K., and D.N.S.

We declare no competing interests.

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
