## [Reviewer comments · Microbiology Spectrum]

Microbiology Spectrum

Co-immunization with pre-erythrocytic antigens alongside circumsporozoite protein can enhance sterile protection against *Plasmodium* sporozoite infection.

Vladimir Vigdorovich, Hardik Patel, Alex Watson, Andrew Raappana, Laura Reynolds, William Selman, Suzannah Beeman, Paul Edlefsen, Stefan Kappe, and D. Sather

Corresponding Author(s): D. Sather, Seattle Children's Research Institute

Review Timeline:

Submission Date:	October 12, 2022
Editorial Decision:	November 23, 2022
Revision Received:	January 11, 2023
Accepted:	February 10, 2023

Editor: Gemma Moncunill

Reviewer(s): The reviewers have opted to remain anonymous.

Transaction Report:

DOI: <https://doi.org/10.1128/spectrum.03791-22>

November 23, 2022

Dr. D. Noah Sather
Seattle Children's Research Institute
Seattle, WA

Re: Spectrum03791-22 (Co-immunization with pre-erythrocytic antigens alongside circumsporozoite protein can enhance sterile protection against *Plasmodium* sporozoite infection.)

Dear Dr. D. Noah Sather:

Thank you for submitting your manuscript to Microbiology Spectrum. Your manuscript has been peer-reviewed, and the comments of the expert reviewers are provided below. When submitting the revised version of your paper, please provide (1) point-by-point responses to the issues raised by the reviewers as file type "Response to Reviewers," not in your cover letter, and (2) a PDF file that indicates the changes from the original submission (by highlighting or underlining the changes) as file type "Marked Up Manuscript - For Review Only". Please use this link to submit your revised manuscript - we strongly recommend that you submit your paper within the next 60 days or reach out to me. Detailed instructions on submitting your revised paper are below.

Link Not Available

Sincerely,

Gemma Moncunill

Journals Department
Reviewer comments:

Reviewer #1 (Comments for the Author):

The study presented in this manuscript focuses on improving the efficacy of malaria vaccine(s). Specifically, the design of the experimental approaches relates to testing of a series of pre-erythrocytic stage *Plasmodium yoelii* sporozoite antigens that combined with Py CSP can enhance the efficacy malaria vaccine in a murine protection model. The experiments are performed with vigor and the results support the claim that a co-immunization with a combination of Py CSP plus another Py sporozoite-stage antigens improves protection induced by CSP alone. There is no doubt that additional pre-erythrocytic *Plasmodium* antigens need to be identified, characterized, and included into a pool of the existing putative vaccines to buttress either RTS,S or similar vaccines based on *P. falciparum* CSP. Several murine studies showing enhancing protection induced by CSP combined with other pre-erythrocytic antigens have already been published.

Amongst the group of the Py sporozoite associated antigens, the investigators found 4 (including TRAP and CelTOS) that enhanced protection in comparison to protection induced by Py CSP alone. The level of protection was evaluated by either a reduction of liver parasite burden or sterile protection against sporozoite infection delivered by a bite from Py infected mosquitoes. The experiments also include quantification by ELISA of antibody titers induced with each (adjuvanted) immunogen against the respective antigens and Py sporozoites. Protective efficacies particularly of CelTOS and TRAP as well immune responses generated by these Plasmodium antigens have been reported previously. Therefore, limiting the immune analysis to antibody titers (as is done in this study) does not provide any tangible mechanisms that may underline enhanced protective mechanism(s) induced with Py CSP + TRAP or Py CSP+ CelTOS. Without additional results that might provide some explanation relating to either the role of antibodies or cellular responses as potential contributors to enhanced protection, the current study appears preliminary. Consequently, the study - as is - does not significantly advance the current understanding of the benefits of combining Plasmodium antigens. The authors themselves discuss that examining antibody titers alone is insufficient to understand the contributions of combining other antigens with CSP. They specifically comment on the necessity to conduct analysis of functional aspects of antibodies as well as their characteristics, e.g., Fc isotypes, and possibly cellular responses in order to arrive at some association between immune responses and protective immunity. Another omission in the experimental design is the absence of mouse groups that would have been co-immunized with CSP plus 2 or 3 other antigens, e.g., CSP+TRAP+CelTOS to examine if protection could exceed 50%.

Reviewer #2 (Comments for the Author):

The article, entitled, "Co-immunization with pre-erythrocytic antigens alongside circumsporozoite protein can enhance sterile protection against Plasmodium sporozoite infection", provides some evidence that combining two malarial antigens can lead to synergy in protective responses. The authors identify and evaluate 8 pre-erythrocytic targets, a few of which have been identified and evaluated as single targets, over the last 2 decades, as potential second targets for a combination approach. The argument for developing approaches to combine antigens has been slow to be realized, as researchers investigate single targets, platforms for delivery and attempt to identify immune mechanisms of protection. To this end, then this study provides some insights in combining PE antigens with CSP, albeit in the *P. yoelii* rodent malaria model.

The targets are expressed in HEK293 cells and purified using affinity chromatography. Final proteins are formulated with an experimental adjuvant 20% Adjuvax which is a biodegradable matrix of carbomer homopolymer and nanoliposomes derived from soy- lecithin, notably this adjuvant stimulates robust antibodies which the characterization of is the primary focus of this paper. The methods used for the evaluation of antibody responses and protection are robust and have been cited from previous work. Minor comments or suggestions for further expansion.

- 1) Can the authors expand in Discussion on the use of Adjuvax and how it compares to other adjuvants in preclinical studies?
- 2) A 15-mosquito bite challenge is quite stringent; do the authors assess for the bolus of sporozoites that are delivered, i.e., feeding efficiency or success?
- 3) Minor: immunization route was not specified in the Materials and Methods.
- 4) Can the authors discuss/justify the selection of Balb/c strain of mice in PyCSP rodent model since much of the work found in the literature is in C57BL6, particularly for *P. falciparum* CSP.

Staff Comments:

Preparing Revision Guidelines

Please return the manuscript within 60 days; if you cannot complete the modification within this time period, please contact me. If you do not wish to modify the manuscript and prefer to submit it to another journal, please notify me of your decision immediately so that the manuscript may be formally withdrawn from consideration by Microbiology Spectrum.

Responses to reviewers' comments

Reviewer 1

“Protective efficacies particularly of CelTOS and TRAP as well immune responses generated by these Plasmodium antigens have been reported previously.”

It is true that CelTOS and TRAP have both undergone clinical development as single vaccine antigens and have not been shown to result in protection strong enough to warrant further development as standalone vaccines. However, the main focus of this study is on co-immunization outcomes and not on immune responses to single-vaccine antigens. To our knowledge, only the TRAP protein has been tested in combination with CSP (as RTS,S and not full CSP, as done here). This is discussed extensively in the Discussion section, and we have noted it in the introduction. One recent co-immunization study has been reported (and is cited in our manuscript), but those vaccines had a vectored “boost” component and evaluated liver-stage antigens, whereas our study focuses on recombinant antigens expressed in sporozoites and on induction of antibodies, and there is no overlap between the antigens tested in that study and what we report here. The present study assesses co-immunization of recombinant protein pre-erythrocytic antigens with CSP to determine whether these antigens can enhance CSP-elicited protection. Thus, this study provides novel information that suggests that co-immunization of recombinant PE antigens may be one way to increase the protective efficacy of CSP-only vaccines.

“Therefore, limiting the immune analysis to antibody titers (as is done in this study) does not provide any tangible mechanisms that may underline enhanced protective mechanism(s) induced with Py CSP + TRAP or Py CSP+ CelTOS. Without additional results that might provide some explanation relating to either the role of antibodies or cellular responses as potential contributors to enhanced protection, the current study appears preliminary. Consequently, the study - as is - does not significantly advance the current understanding of the benefits of combining Plasmodium antigens. The authors themselves discuss that examining antibody titers alone is insufficient to understand the contributions of combining other antigens with CSP. They specifically comment on the necessity to conduct analysis of functional aspects of antibodies as well as their characteristics, e.g., Fc isotypes, and possibly cellular responses in order to arrive at some association between immune responses and protective immunity.”

We agree that defining the mechanisms of protection, or features within the antibody response associated with protection, are the logical next steps for this work. However, the goal of the current study was to survey protective responses elicited by several protein targets expressed in sporozoites and identify those that enhance CSP-mediated sterile protection. Although these findings don't offer significant mechanistic insights into the nature of this protection enhancement, they demonstrate two important things: (1) immunization with another protein alongside CSP doesn't appear to diminish

CSP-mediated protection and (2) even targets that elicit no discernable protection on their own (e.g., HSP70-2 or P52) can enhance that of CSP. We hope that our discussion surrounding antibody titers was not misconstrued. We did not intend to imply that antibodies are not the effector responsible for protection. Rather, we intended to make the point that antibody binding titers alone can't explain the differences in protection, and that a more detailed analysis of the specific features of the antibody response would likely help to identify advantageous features associated with protection (e.g., avidity, IgG subclass, etc). This line of inquiry is of significant interest to us, and will be a major focus of our follow up studies. We have clarified some of the language surrounding this, and more clearly discuss the scope of this current study and its future directions.

“Another omission in the experimental design is the absence of mouse groups that would have been co-immunized with CSP plus 2 or 3 other antigens, e.g., CSP+TRAP+CeITOS to examine if protection could exceed 50%.”

We agree that a rigorous examination of combinations of three (or more) total immunogens is the logical next experiment, and it is a feature of our future plans on this project. However, to conduct such studies for this manuscript would fall outside of the scope of this pilot study. Combinatorial experiments with all of our antigens in a CSP+2 PE antigens format would require a minimum of 70 groups of 5 mice (>350 mice) with repetition and controls. Even if restricted to those antigens that show increased protection here, the study would require a minimum of 20 groups of mice (~100 mice). Thus, although we fully agree with the reviewer, we cannot conduct those studies for this manuscript.

Reviewer 2

Minor comments or suggestions for further expansion.

1) Can the authors expand in Discussion on the use of Adjuplex and how it compares to other adjuvants in preclinical studies?

We have added the requested information on the preclinical use of Adjuplex, with citations. Adjuplex is a carbomer based nano-particle adjuvant. Although Adjuplex has not been tested in human trials, it has been used in numerous non-human primate and other preclinical animal models as an adjuvant against a broad array of pathogens, including viruses, fungi, and parasites. Further, carbomer class adjuvants are an area of intense development as next generation adjuvants, as they are biocompatible, non-toxic, and elicit broad immunity, and they are already commonly used as formulation components and stabilizers in pharmaceuticals for human use.

2) A 15-mosquito bite challenge is quite stringent; do the authors assess for the bolus of sporozoites that are delivered, i.e., feeding efficiency or success?

Our previous studies have shown that 15 mosquito bites provides the most optimal balance between infection efficiency and challenge stringency in the context of protection offered by PyCSP immunization. Here we re-optimized the challenge dose with our current vaccine regimen, and confirmed that 15 mosquito bites provided the most reproducible rates of infection, as illustrated by the infection efficiency in our control groups. To ensure rigor and reproducibility, before initiating challenge experiments, every batch of infected mosquitoes was screened for the prevalence of infection (by checking mosquito midguts) and only those mosquito cages that showed >80 % prevalence were used for the challenge studies. In addition, we counted the sporozoite load per mosquito by harvesting the salivary glands on day-14 post infected blood meal, i.e., before performing the bite-challenge experiments on day-15, to verify that the infection level exceeded our cutoff of 10,000 sporozoites/mosquitoes. After the challenge, we verified that the mosquitoes had taken a blood meal by visual assessment. We have updated this in the Materials and Methods section.

3) Minor: immunization route was not specified in the Materials and Methods.

The animals were immunized intramuscularly. We have added this information to the Materials and Methods.

4) Can the authors discuss/justify the selection of Balb/c strain of mice in PyCSP rodent model since much of the work found in the literature is in C57BL6, particularly for P. falciparum CSP.

Our choice of strain was due to the use of our *P. yoelii* mosquito bite challenge model that was developed and validated in house and utilizes BALB/c mice. This model has been

optimized for live immunization and mosquito bite challenge, which provides the natural route of infection and sporozoite transit through the tissues where antibodies are thought to be the most functional. Further, we have reported that BALB/c mice are more susceptible to *Py* infection (Kaushansky et al., 2015; Sack et al., 2014), making it the most appropriate host for studies using *Py*, and have recently reported several studies on antibody inhibition against sporozoite invasion using this model (Vijayan et al., 2021; Visweswaran et al., 2022; Wilder et al., 2022). The reviewer is correct that a large body of work exists with C57BL6, especially with the *P. berghei* model, but there is also a large body of work with BALB/c mice, and we believe our use of this strain is not unusual. We have provided these citations and updated language in the manuscript to better justify the use of this model.

References for the above response:

- Kaushansky, A., Austin, L. S., Mikolajczak, S. A., Lo, F. Y., Miller, J. L., Douglass, A. N., Arang, N., Vaughan, A. M., Gardner, M. J., & Kappe, S. H. I. (2015). Susceptibility to *Plasmodium yoelii* preerythrocytic infection in BALB/c substrains is determined at the point of hepatocyte invasion. *Infection and Immunity*, *83*(1), 39–47.
- Sack, B. K., Miller, J. L., Vaughan, A. M., Douglass, A., Kaushansky, A., Mikolajczak, S., Coppi, A., Gonzalez-Aseguinolaza, G., Tsuji, M., Zavala, F., Sinnis, P., & Kappe, S. H. I. (2014). Model for in vivo assessment of humoral protection against malaria sporozoite challenge by passive transfer of monoclonal antibodies and immune serum. *Infection and Immunity*, *82*(2), 808–817.
- Vijayan, K., Visweswaran, G. R. R., Chandrasekaran, R., Trakhimets, O., Brown, S. L., Watson, A., Zuck, M., Dambrauskas, N., Raappana, A., Carbonetti, S., Kelnhofner-Millevolte, L., Glennon, E. K., Postiglione, R., Sather, D. N., & Kaushansky, A. (2021). Antibody interference by a non-neutralizing antibody abrogates humoral protection against *Plasmodium yoelii* liver stage. *Cell Reports*, *36*(5), 109489.
- Visweswaran, G. R. R., Vijayan, K., Chandrasekaran, R., Trakhimets, O., Brown, S. L., Vigdorovich, V., Yang, A., Raappana, A., Watson, A., Selman, W., Zuck, M., Dambrauskas, N., Kaushansky, A., & Noah Sather, D. (2022). Germinal center activity and B cell maturation are associated with protective antibody responses against *Plasmodium* pre-erythrocytic infection. In *PLOS Pathogens* (Vol. 18, Issue 7, p. e1010671). <https://doi.org/10.1371/journal.ppat.1010671>
- Wilder, B. K., Vigdorovich, V., Carbonetti, S., Minkah, N., Hertoghs, N., Raappana, A., Cardamone, H., Oliver, B. G., Trakhimets, O., Kumar, S., Dambrauskas, N., Arredondo, S. A., Camargo, N., Seilie, A. M., Murphy, S. C., Kappe, S. H. I., & Sather, D. N. (2022). Anti-TRAP/SSP2 monoclonal antibodies can inhibit sporozoite infection and may enhance protection of anti-CSP monoclonal antibodies. *Npj Vaccines*, *7*(1), 1–13.

February 10, 2023

Dr. D. Noah Sather
Seattle Children's Research Institute
Seattle, WA

Re: Spectrum03791-22R1 (Co-immunization with pre-erythrocytic antigens alongside circumsporozoite protein can enhance sterile protection against *Plasmodium* sporozoite infection.)

Dear Dr. D. Noah Sather:

Your manuscript has been accepted, and I am forwarding it to the ASM Journals Department for publication. You will be notified when your proofs are ready to be viewed.

Sincerely,

Gemma Moncunill
Editor, Microbiology Spectrum
